# Results of PD-L1 Analysis of Women Treated with Durvalumab in Advanced Endometrial Carcinoma (PHAEDRA)

**DOI:** 10.3390/cancers15010254

**Published:** 2022-12-30

**Authors:** Deborah Smith, Kristy P. Robledo, Sonia Yip, Michelle M. Cummins, Peey-Sei Kok, Yeh Chen Lee, Michael Friedlander, Sally Baron-Hay, Catherine Shannon, Jermaine Coward, Philip Beale, Geraldine Goss, Tarek Meniawy, Janine Lombard, Amanda B. Spurdle, John Andrews, Martin R. Stockler, Linda Mileshkin, Yoland Antill

**Affiliations:** 1Mater Pathology, Mater Research, University of Queensland, Brisbane, QLD 4072, Australia; 2National Health and Medical Research Council Clinical Trials Centre, University of Sydney, Camperdown, Sydney, NSW 2050, Australia; 3Prince of Wales Clinical School, University of New South Wales and Royal Hospital for Women, Randwick, Sydney, NSW 2031, Australia; 4Chris O’Brien Lifehouse, Camperdown, Sydney, NSW 2050, Australia; 5Royal North Shore Hospital, St Leonards, NSW 2065, Australia; 6Mater Cancer Centre, South Brisbane, QLD 4101, Australia; 7ICON Cancer Centre, South Brisbane, QLD 4101, Australia; 8Monash Medical Centre, Melbourne, VIC 3168, Australia; 9Sir Charles Gairdner Hospital, Perth, WA 6009, Australia; 10Calvary Mater Newcastle, Newcastle, NSW 2298, Australia; 11QIMR Berghofer Medical Research Institute, Herston, QLD 4006, Australia; 12Peter MacCallum Cancer Centre, Melbourne, VIC 3000, Australia; 13Cabrini Health, Malvern, Melbourne, VIC 3144, Australia; 14Faculty of Medicine, Nursing and Health Sciences, Monash University, Clayton, VIC 3800, Australia

**Keywords:** programmed death ligand 1, immune checkpoint inhibitors, endometrial carcinoma, immunohistochemistry, receiver operating characteristic analyses

## Abstract

**Simple Summary:**

Until recently the outcome for women with advanced endometrial carcinoma has been poor. New immune therapies have resulted in much better outcomes for some, but not all, women with advanced disease. Determining which women are likely to respond is important, both to identify potential responders and prevent overtreatment of women who are not likely to respond. In this study we looked to see if there were any other markers that could be used to better predict those tumors that would be more likely to respond to the immune therapy known as durvalumab versus those tumors more likely to be resistant. Using statistical methods to evaluate each of the components examined we determined the cut-off point with the best performance. We found that the presence of tumor associated inflammatory cells had the strongest association with response. We also found that an algorithm derived from our best performing cut-off points identified women not likely to respond to treatment. While the presence of inflammatory cells was not significant when mismatch repair status was considered, our novel algorithm was. This is a small study and the findings do require validation in a larger group of women.

**Abstract:**

Women with advanced endometrial carcinoma (EC) with mismatch repair (MMR) deficiency have improved outcomes when treated with immune checkpoint inhibitors; however, additional biomarkers are needed to identify women most likely to respond. Scores for programmed death ligand 1 (PD-L1), immunohistochemical staining of tumor (TC+), immune cells (IC+) and presence of tumor-associated immune cells (ICP) on MMR deficient (*n* = 34) and proficient (*n* = 33) EC from women treated with durvalumab in the PHAEDRA trial (ANZGOG1601/CTC0144) (trial registration number ACTRN12617000106336, prospectively registered 19 January 2017) are reported and correlated with outcome. Receiver operating characteristic (ROC) analyses and area under the ROC curve were used to determine optimal cutpoints. Performance was compared with median cutpoints and two algorithms; a novel algorithm derived from optimal cutpoints (TC+ ≥ 1 or ICP ≥ 10 or IC+ ≥ 35) and the Ventana urothelial carcinoma (UC) algorithm (either TC+ ≥ 25, ICP > 1 and IC+ ≥ 25 or ICP = 1 and IC+ = 100). The cutpoint ICP ≥ 10 had highest sensitivity (53%) and specificity (82%), being prognostic for progression-free survival (PFS) (*p* = 0.01), while the optimal cutpoints algorithm was associated with overall survival (*p* = 0.02); these results were not significant after adjusting for MMR status. The optimal cutpoints algorithm identified non-responders (*p* = 0.02) with high sensitivity (88%) and negative predictive value (92%), remaining significant after adjustment for MMR. Although MMR status had the strongest association with response, further work to determine the significance of ICP ≥ 10 and the novel optimal cutpoint algorithm is needed.

## 1. Introduction

Advanced and recurrent endometrial carcinoma (AEC) has a high mortality rate, with little improvement in therapeutic options over the past several decades. Recently treatment with novel immune checkpoint inhibitors (ICI) has resulted in improved outcomes, particularly for carcinomas with mismatch repair deficiency (dMMR) [1,2,3,4]. Currently four molecular subtypes of EC are recognized that are complementary to the traditional FIGO system for histotyping and grading [5]. These are: (1) polymerase-epsilon (*POLE*) ultra-mutated, (2) microsatellite instability (MSI), (3) copy number high with p53 mutational status (p53) and (4) copy number low, microsatellite stable with a non-specific molecular profile (NSMP) [6]. ICIs are theoretically more likely to invoke good clinical response in malignancies with high tumor mutational rates [7]; of the EC molecular subtypes both *POLE* and MSI-H carcinomas are known to have high mutation rates. Mismatch repair deficient (dMMR) EC can be identified using immunohistochemistry (IHC), which is cheap, widely available in the Australian setting, and generally associated with MSI-H. Identifying other subgroups with high mutational rates is more complex, with molecular tests for *POLE* status and high tumor mutation burden (hTMB) more expensive and limited to specialized laboratories.

Biomarkers to further optimize patient selection for ICIs by identifying the women most likely to respond, that is, prognostic biomarkers, remains an unmet need. Programmed death ligand 1 (PD-L1) IHC is attractive because of its relatively widespread availability and clinical experience in other malignancies, such as non-small cell lung carcinoma (NSCLC). Unlike malignancies such as NSCLC, where reporting guidelines and treatment cut-offs are established, currently there is no validated scoring algorithm for assessment of PD-L1 status in EC [3,4,8,9]. In addition, studies examining the relationship between EC PD-L1 status and ICI clinical response are limited, with inconsistent results reported to date [3,4,10]. Lastly, the use of cutpoints enables potentially complex relationships over outcomes to be simplified and minimizes the impact of inter-observer reliability.

PHAEDRA is an Australia New Zealand Gynaecological Oncology Group (ANZGOG)-led, non-comparative, single-arm phase 2 trial that assessed the activity of durvalumab, an anti-PD-L1 monoclonal antibody, in both dMMR and proficient MMR (pMMR) cohorts of women with AEC. The objective tumor response rate (OTRR) in the dMMR cohort was 47% (17/36; 95% CI 32–63) and 3% in the pMMR cohort (1/35, 95% CI: 1–15) [1]. The analyses reported here explores interobserver reliability of tumor expression of PD-L1 and identifies optimal cutpoints for tumor expression of PD-L1 and whether it is a prognostic biomarker for clinical outcomes of women participating in the PHAEDRA study.

## 2. Materials and Methods

### 2.1. Study Population

PHAEDRA included two AEC cohorts (dMMR and pMMR) based on site-determined MMR status using IHC. This has been previously reported [1]. All women received intravenous durvalumab 1500 mg 4-weekly until disease progression, prohibitive toxicity, or withdrawal from the study. Tumor tissue less than 18 months old or repeat biopsy prior to study enrollment was required. Central review of MMR stains at Mater Pathology, Brisbane, Australia, was undertaken retrospectively following enrollment; where original slides were unavailable these were repeated [1].

### 2.2. PD-L1 Immunohistochemistry

Formalin-fixed paraffin embedded tissue blocks or unstained tissue sections on charged slides (Trajan Scientific and Medical, Melbourne, Australia) were stained for PD-L1 protein expression using IHC at a central laboratory (Mater Pathology). All specimens were stained within 4 weeks of sectioning; staining of whole slides was performed by the VENTANA PD-L1 (SP263) Assay (Ventana 741-4905, Roche Diagnostics, Mannheim, Germany) and visualized using a Benchmark Ultra IHC/ISC system (Roche Diagnostics, Indianapolis, IN, USA), per the manufacturer’s instructions, and using appropriate positive controls. The full PD-L1 staining protocol is included with the Appendix A. Matching haematoxylin and eosin (H&E) sections were also obtained.

### 2.3. PD-L1 Scoring

Scoring of PD-L1 stained whole tissue sections was performed by two specialist anatomical pathologists with training and experience reporting PD-L1 stains (Drs Smith and Snell), and reported according to clinical guidelines. Scoring was performed independently following an initial training set of ten cases, with the scores determined by each pathologist for the three components then averaged. Specimens with less than 100 viable TC were excluded.

The Ventana PD-L1 assay interpretation guidelines for UC [11] was chosen as this method scores tumor and IC separately, and the scoring system is also validated for use with the SP263 clone, including treatment with durvalumab [9,12]. Components scored were: (1) the percentage of TC with partial or complete membranous staining for PD-L1 (TC+); (2) tumor-associated IC, determined as a percentage of the tumor area including all IC within the tumor reactive stroma, between tumor islands, and within the tumor proper (ICP); and (3) IC with any staining for PD-L1, granular or cytoplasmic, expressed as a percentage of IC present (IC+). Each of the components (TC+, ICP and IC+) were scored in deciles and quartiles; ICP was also scored at 1 and 5. If ICP was <1 an IC+ score of 100 was required to be considered PD-L1 positive. Examples of PD-L1 staining and the scores as determined by each pathologist are given in Appendix A.

### 2.4. Statistical Analyses

Inter-observer reliability for scoring of PD-L1 stained whole tissue sections was performed using Bland–Altman techniques for each of the individual components (TC+, ICP and IC+). Receiver operating characteristic (ROC) analyses were performed for each individual component to predict the OTRR, defined as either a partial response or a complete response according to iRECIST 1.1 [13]. The area under the ROC curve (AUC), sensitivity and specificity were calculated to provide a summary measure of predictive performance. Optimal cutpoints were selected using the point closest to the top left of each plot. The urothelial carcinoma (UC) algorithm, previously validated for clone SP263 (either TC+ ≥ 25%, ICP > 1 and IC+ ≥ 25% or ICP = 1 and IC+ = 100) [11] was compared to an algorithm derived from the combination of the optimal cutpoints (OC; TC+ ≥ 1, ICP ≥ 10 or IC+ 35). The prognostic value of the proposed optimal cutpoint for PD-L1 staining in EC was assessed with progression-free survival (PFS) according to iRECIST and overall survival (OS) using Kaplan–Meier curves, and adjusted analyses using Cox proportional hazards regression. OTRR was assessed using logistic regression.

## 3. Results

### 3.1. Baseline Characteristics

The study population included 71 women with AEC, 36 with dMMR, and 35 pMMR [1]. Sixty-seven women had sufficient tumor for PD-L1 testing (33 pMMR, 34 dMMR) and were eligible for this analysis. Tissue for PD-L1 evaluation included both uterine primary (*n*=27) and metastatic sites (*n*=40). The median age was 67 years (IQR: 60–72); the majority of tumor histology was endometrioid (76%, 51/67) or serous (16%, 11/67) subtypes (Appendix A).

### 3.2. Immune Cell and PD-L1 Results

Figure 1 explores inter-observer reliability for TC+, ICP and IC+, demonstrating increasing variability for each as the values increase. Generally, one observer appeared to score higher than the other observer for ICP and IC+ measurements.

To predict OTRR, AUC were 0.667 (95% CI: 0.512–0.821), 0.726 (95% CI: 0.577–0.874) and 0.644 (95% CI: 0.492–0.797) for TC+, ICP and IC+, respectively (Figure 2). Plots of the distributions of these components by responders and non-responders is given in Appendix A. Optimal cutpoints were determined as TC+ ≥ 1, ICP ≥ 10 and IC+ ≥ 35. Predictive performance of optimal cutpoints was compared to the median cutpoints and two algorithms: 1) an optimal cutpoint (OC) algorithm (TC+ ≥ 1 or ICP ≥ 10 or IC+ ≥ 3) and 2) the Ventana UC algorithm (either TC+ ≥ 25, ICP > 1 and IC+ ≥ 25 or ICP = 1 and IC+ = 100) [11], (Table 1).

The observed OTRR were 33% vs. 20% in participants with TC+ ≥ 1 vs. < 1; 50% vs. 16% for ICP ≥ 10 vs. ICP < 10, and 31% vs. 24% for IC+ ≥ 35, vs. IC+ < 35 (Table 1). ICP ≥ 10 was the cutpoint with the highest sensitivity (53%) and specificity (82%), and in univariate analyses ICP alone was prognostic for OTRR (*p* = 0.007). However, when adjusted for MMR status, ICP ≥ 10 was not prognostic for OTRR (*p* = 0.12, Appendix A). The OC algorithm identified non-responders (*p* = 0.02) with high sensitivity (88%) and negative predictive value (92%), but low specificity (48%) and positive predictive value (37%). This remained prognostic for OTRR after adjustment for MMR status (*p* = 0.035).

ICP ≥ 10 was found to be prognostic for PFS (logrank *p* = 0.01), while TC+ (*p* = 0.25), IC+ (*p* = 0.48) and the UC algorithm (*p* = 0.08) were not (Figure 3). In a model adjusting for MMR status and ICP ≥ 10, pMMR was associated with poorer PFS (HR for pMMR 2.99, 95% CI: 1.61–5.57, *p* < 0.001), and ICP ≥ 10 was no longer associated with PFS (HR for ICP ≥ 10 0.59, 95% CI 0.28–1.23, *p* = 0.16). None of the individual PD-L1 cutpoints were prognostic for OS (TC+ *p* = 0.18, ICP *p* = 0.07 or IC+ *p* = 0.23, Figure 4). Comparisons of univariate and adjusted models for PFS and OS are given in Appendix A. The UC algorithm was associated with OS (logrank *p* = 0.02), but not after adjustment for MMR status (HR for UC algorithm: 0.53, 95% CI: 0.25–1.12, *p* = 0.10). While the UC algorithm was not prognostic for OS (*p* = 0.063, Figure 4E), it was for PFS (*p* = 0.013, Figure 3E); however, it did not remain prognostic after adjustment for MMR status (*p* = 0.35).

## 4. Discussion

Here, we present the exploratory analyses of the prognostic ability of PD-L1 expression with OTRR, PFS and OS in women treated with durvalumab for AEC. Compared to TC+ and IC+ optimal scores, ICP ≥ 10 had the highest sensitivity and specificity and was prognostic for OTRR and PFS, but not OS. These associations were no longer evident after adjusting for MMR status. However, the novel OC algorithm developed here detected patients that are unlikely to respond to treatment (TC+ < 1 and ICP < 10 and IC+ < 35), even after adjustment for MMR status.

Reported PD-L1 expression in unselected EC ranges from 1% to 44% [8,14], with higher rates in dMMR tumors (26% to 48%) [7,8]. Variations in antibody clone, assessment method and cutpoint makes comparisons between studies difficult [3,10,12,15]. The prognostic significance of PD-L1 expression is complex, with some studies reporting longer PFS and OS associated with high PD-L1 expression [7,16], but poorer survival associated with immune cell PD-L1 expression [16,17]. No association with survival was found in other studies [14]. However, a recent meta-analysis including retrospective studies comprising 1615 patients found no association between PD-L1 status and survival; high PD-L1 correlating only with poor differentiation and high tumor stage [18]. Furthermore, to date there has been no conclusive demonstration of associations between PD-L1 status and outcomes of ICI treatment for EC [9,10,11]. This contrasts with other malignancies, where high PD-L1 is predictive of treatment response and may be a prerequisite for treatment [11].

Cancers with dMMR typically have an MSI high phenotype, resulting in high mutational frequency and higher production of novel frameshift peptide antigens. The abundance and “foreign” nature of these neo-antigens likely explains the strong CD3+ and CD8+ T-cell responses, which are associated with higher response rates to ICIs. Longer PFS was observed in a range of solid malignancies with both high levels of T-cell inflammatory gene expression profiles and hTMB in response to pembrolizumab [10]. In contrast, neither TMB nor tumor infiltrating lymphocytes correlated with response in a study of avelumab in EC [3]. While MMR status remained the strongest prognostic variable for outcomes similar to previous studies [3,19], our findings suggest that the OC algorithm may identify those women less likely to respond.

Recent work by Willvonseder et al. [20], who examined the immune microenvironment in the context of molecularly characterized EC, found a subset of high grade (FIGO grade 3) NSMP (pMMR/*POLE* wildtype) ECs also had T-cell inflamed stroma. This potentially represents an additional immunogenic subgroup [20], which may be associated with hTMB and response to ICIs, the significance of which requires further study.

It is important to recognize the limitations of this study. These were post hoc analyses, and samples included both primary tumors and biopsies from metastases, with insufficient numbers within each of these subsets for further analysis. Additionally, we are yet to assess other molecular correlates that may affect PD-L1 expression, such as mechanism of dMMR. Finally, as a single-arm study it is not possible to determine if any association of efficacy is predictive.

A strength of our study is the use of whole tissue sections rather than tissue microarray cores; in our experience both PD-L1 staining and distribution of inflammatory cells are heterogeneous, a finding also noted by others [9,14,20].

## 5. Conclusions

In this exploratory analysis of PD-L1 expression and association with outcome in AEC treated with durvalumab, ICP alone was prognostic for OTRR and PFS, while TC+ and IC+ were not. After the inclusion of MMR status, ICP alone was no longer prognostic. A novel algorithm including ICP, TC+ and IC+ was able to identify non-responders, even after adjustment for MMR status. Further work is needed to validate these findings in a larger cohort.

## Figures and Tables

**Figure 1 cancers-15-00254-f001:**
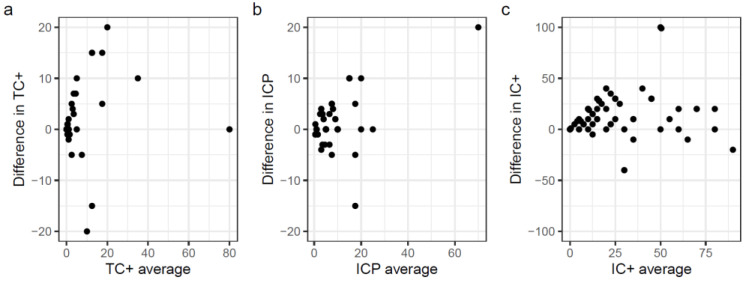
Bland–Altman plots for the three components scored. (**a**) TC+, (**b**) ICP and (**c**) IC+.

**Figure 2 cancers-15-00254-f002:**
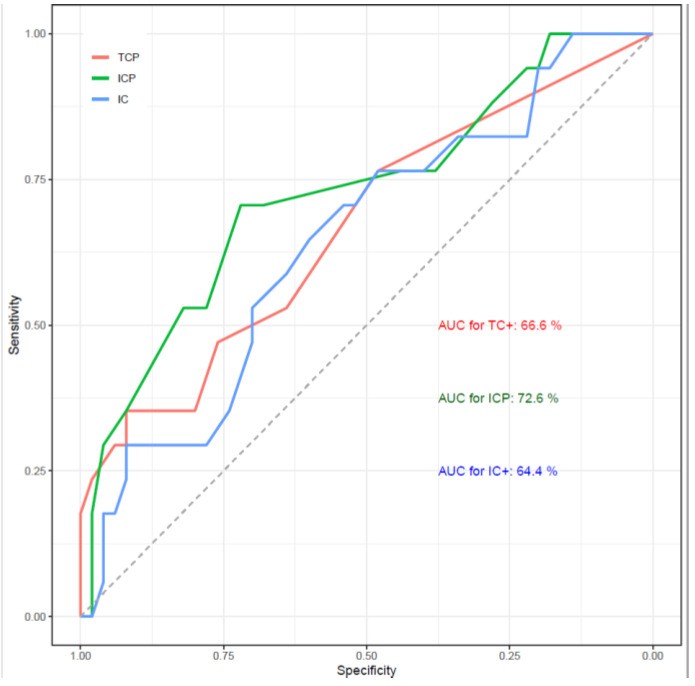
Area under the curve for best overall response for TC+ (red), ICP (green) and IC+ (blue).

**Figure 3 cancers-15-00254-f003:**
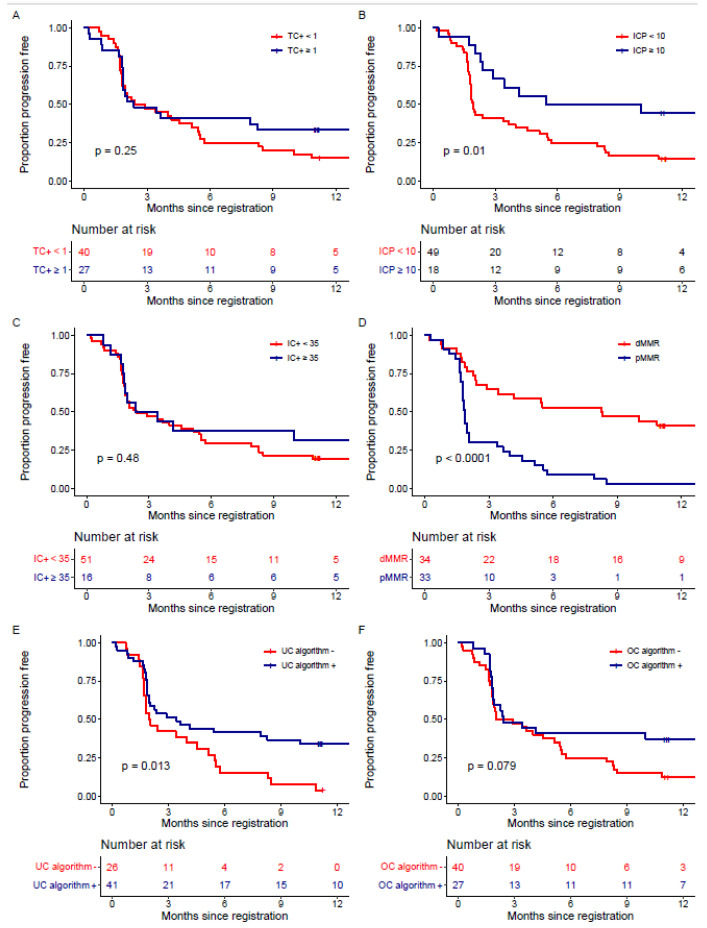
Kaplan–Meier curves for progression-free survival for (**A**) TC+, (**B**) ICP, (**C**) IC+, (**D**) MMR, (**E**) the urothelial cancer (UC) algorithm and (**F**) the optimal cutpoint (OC) algorithm.

**Figure 4 cancers-15-00254-f004:**
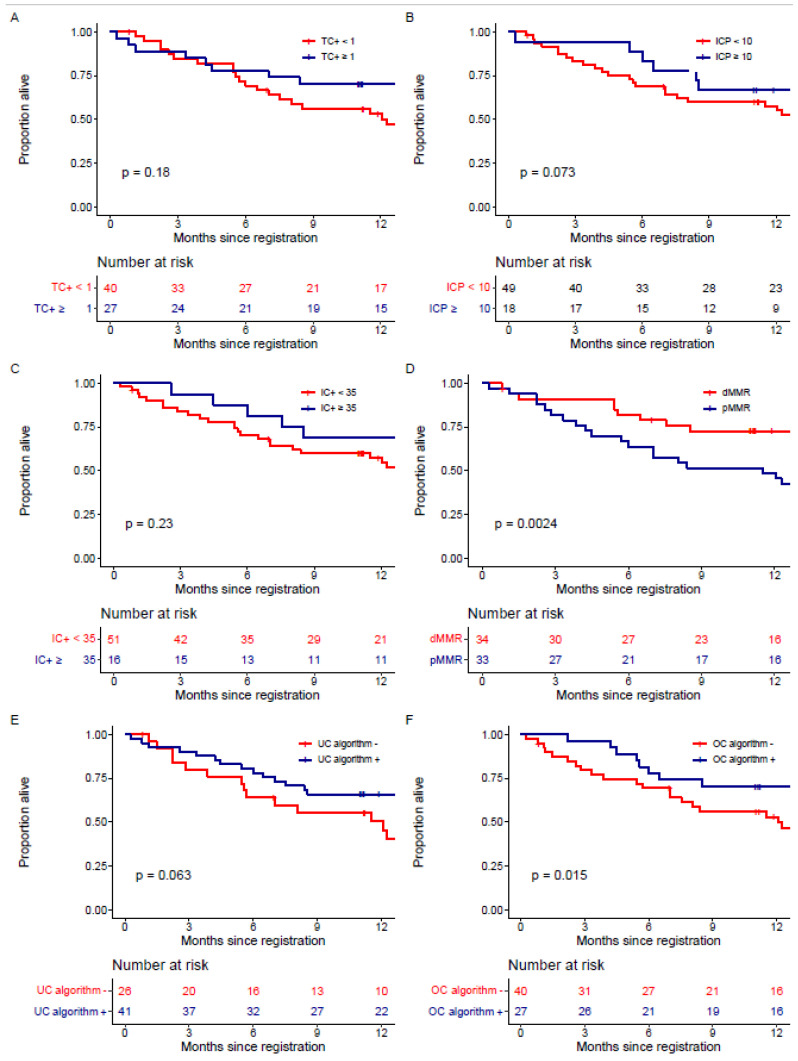
Kaplan–Meier curves for overall survival for (**A**) TC+, (**B**) ICP, (**C**) IC+, (**D**) MMR, (**E**) the urothelial cancer (UC) algorithm and (**F**) the optimal cutpoint (OC) algorithm.

**Table 1 cancers-15-00254-t001:** Associations between OTRR and the three components and the algorithms.

PD-L1 Score	Responses Below Cutpoint * (n/N (%))	Responses above Cutpoint * (n/N (%))	Positive Predictive Value (95% CI)	Negative Predictive Value (95% CI)	Sensitivity (95% CI)	Specificity (95% CI)
Optimal cutpoints						
○ TC+ ≥ 1	8/40 (20%)	9/27 (33%)	33% (17%, 54%)	80% (64%, 91%)	53% (28%, 77%)	64% (49%, 77%)
○ ICP ≥ 10	8/49 (16%)	9/18 (50%)	50% (26%, 74%)	84% (70%, 93%)	53% (28%, 77%)	82% (69%, 91%)
○ IC+ ≥ 35	12/51 (24%)	5/16 (31%)	31% (11%, 59%)	76% (63%, 87%)	29% (10%, 56%)	78% (64%, 88%)
Median cutpoints						
○ TC+ ≥ 1	8/40 (20%)	9/27 (33%)	33% (17%, 54%)	80% (64%, 91%)	53% (28%, 77%)	64% (49%, 77%)
○ ICP ≥ 5	4/26 (15%)	13/41 (32%)	32% (18%, 48%)	85% (65%, 96%)	76% (50%, 93%)	44% (30%, 59%)
○ IC+ ≥ 20	5/32 (16%)	12/35 (34%)	34% (19%, 52%)	84% (67%, 95%)	71% (44%, 90%)	54% (39%, 68%)
Other proposed cutpoints						
UC algorithm †	6/40 (15%)	11/27 (41%)	41% (22%, 61%)	85% (70%, 94%)	65% (38%, 86%)	68% (53%, 80%)
OC algorithm ‡	2/26 (8%)	15/41 (37%)	37% (22%, 53%)	92% (75%, 99%)	88% (64%, 99%)	48% (34%, 63%)

TC+: tumor cells with positive staining, ICP: percentage of tumor area occupied by immune cells, IC+: percentage of tumor-associated immune cells with positive staining. *: Within all patients above or below the cutpoint. † Urothelial cancer (UC) algorithm: TC+ ≥ 25% OR ICP > 1 & IC+ ≥ 25 OR ICP = 1 & IC+ = 100. ‡ Optimal cutpoint (OC) algorithm: TC+ ≥ 1 OR ICP ≥ 10 OR IC+ ≥ 35.

## Data Availability

All data relevant to the study are included in the article or uploaded as Appendix A. Individual deidentified subject data that underlie the results reported in this article will be shared upon reasonable request to the corresponding author, after approval by the PHAEDRA trial management committee. Study protocol has been shared in previous publications.

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
