# Peer review of "Results of PD-L1 Analysis of Women Treated with Durvalumab in Advanced Endometrial Carcinoma (PHAEDRA)"

_cancers, 2022, doi:10.3390/cancers15010254_

Round 1

Reviewer 1 Report

This short paper would meet the remit of the journal and would be of interest to the readership.  The manuscript describes the outcome of an analysis of PDL-1 levels measured by IHC in tumour cells, immune cells and tumour associated cells in mismatach repair deficient and proficient samples was assessed as a biomarker. The study shows no significant prognostic association with PFS, OS when MMR status was considered.  However optimal cutpoint values significantly identified non-responders, which is an interesting result.

The paper is well written and data is presented appropriately.

More detail on the PDL-1 staining protocol would be useful. Figures showing example staining would also be appreciated.

Could separate analyses be performed on primary and secondary tumours, as the immune environment in metastatic niches may differ and also vary between the site of metastasis?

Author Response

Please see the attachment - I cannot attach the photos and will need to find another way to send. Thank-you.

Reviewer 2 Report

The Australia New Zealand Gynecological Oncology Group presents a study of significant potential interest.

Unfortunately, the results are not presented in the most accurate way.

For this reason, I have to underline some points to make the text more consistent with the study and to make the study more reproducible.

1.       Urothelial cancer (UC) algorithm (TC+ ≥ 25% OR ICP>1& IC+ ≥ 25 OR ICP = 1 & IC+ = 100) and  Optimal cutpoint (OC) algorithm (TC+ ≥ 1 OR ICP ≥ 10 OR IC+ ≥ 35) should be described more fully in materials and methods.

2.       The legend of figure 1 is missing and in its place, there is part of the text of the results linked to figure 2. Also, the initial part of this text is evidently missing.

3.       The authors mention Figures 1, 2, 3, and 4 in the figure captions, but refer to Figures I, II, III, and IV in the text.

4.       Where is the Table S1?

5.       On page 5, line 3 and 4 from the top what does the phrase “However, when adjusted for MMR status and ICP ≥ 10, ICP ≥ 10 was not prognostic for OTRR (Table S1)” mean?

6.       In the legend of figure 3 the authors describe the boxes A, B, C, D, E and F, but in the figure the six boxes have no letter.

7.       At the bottom of page 5 the sentence fragment " As compared to TC+ (p = 0.25), IC+ (p = 0.48) and the UC algorithm (p = 0.08) (Figure II)." does not make sense when referring to figure 2 and it does not make sense as a fragment. It would make more sense if related to the sentence immediately preceding figure 3 and referring to figure 3.

8.       In the legend of figure 4 the authors describe the boxes A, B, C, D, E and F, but in the figure the six boxes have no letter.

9.       At the bottom of page 6 the sentence “But not after adjustment for MMR status (HR for UC algorithm: 0.53, 95% CI: 0.25–1.12, p = 0.10). None of the individual PD-L1 cutpoints were prognostic for OS (TC+ p =0.18, ICP p = 0.07 or IC+ p = 0.23). While the OC algorithm was not prognostic for OS (p =0.063), it was for PFS (p = 0.013); however, it did not remain prognostic after adjustment for MMR status (p = 0.35).” is difficult to understand if not appropriately referred to the various figures where the data are reported

10.   On page 7 in the conclusions, the sentence "A novel algorithm including ICP, TC+ and IC+ was able to identify non-responders, even after adjustment for MMR status." is not justified by the results as they are now described.
